# Diversity, taxonomy, and evolution of archaeal viruses of the class *Caudoviricetes*

Ying Liu[1]☯, Tatiana A. Demina[2]☯, Simon Roux[3], Pakorn Aiewsakun[4,5,6], Darius Kazlauskas[7], Peter Simmonds[4], David Prangishvili[1,8], Hanna M. Oksanen[2]*, Mart Krupovic[1]*

**1** Institut Pasteur, Université de Paris, Archaeal Virology Unit, Paris, France, **2** Molecular and Integrative Biosciences Research Programme, Faculty of Biological and Environmental Sciences, University of Helsinki, Helsinki, Finland, **3** DOE Joint Genome Institute, Lawrence Berkeley National Laboratory, Berkeley, California, United States of America, **4** Nuffield Department of Medicine, University of Oxford, Oxford, United Kingdom, **5** Department of Microbiology, Faculty of Science, Mahidol University, Bangkok, Thailand, **6** Pornchai Matangkasombut Center for Microbial Genomics, Department of Microbiology, Faculty of Science, Mahidol University, Bangkok, Thailand, **7** Institute of Biotechnology, Life Sciences Center, Vilnius University, Vilnius, Lithuania, **8** Ivane Javakhishvili Tbilisi State University, Tbilisi, Georgia

☯ These authors contributed equally to this work.

* hanna.oksanen@helsinki.fi (HMO); mart.krupovic@pasteur.fr (MK)

**Data Availability Statement:** All new genome sequences of virus isolates were submitted to GenBank (accession numbers MZ334492-MZ334528). All protein sequences used for phylogenetic analyses can be downloaded from the

## Abstract

The archaeal tailed viruses (arTV), evolutionarily related to tailed double-stranded DNA (dsDNA) bacteriophages of the class *Caudoviricetes*, represent the most common isolates infecting halophilic archaea. Only a handful of these viruses have been genomically characterized, limiting our appreciation of their ecological impacts and evolution. Here, we present 37 new genomes of haloarchaeal tailed virus isolates, more than doubling the current number of sequenced arTVs. Analysis of all 63 available complete genomes of arTVs, which we propose to classify into 14 new families and 3 orders, suggests ancient divergence of archaeal and bacterial tailed viruses and points to an extensive sharing of genes involved in DNA metabolism and counterdefense mechanisms, illuminating common strategies of virus–host interactions with tailed bacteriophages. Coupling of the comparative genomics with the host range analysis on a broad panel of haloarchaeal species uncovered 4 distinct groups of viral tail fiber adhesins controlling the host range expansion. The survey of metagenomes using viral hallmark genes suggests that the global architecture of the arTV community is shaped through recurrent transfers between different biomes, including hypersaline, marine, and anoxic environments.

## Introduction

Bacteriophages with helical tails and icosahedral capsids (tailed bacteriophages), classified into the class *Caudoviricetes* [1], represent the most widespread, abundant, and diverse group of viruses on our planet [2] and are likely to infect hosts from most, if not all, known bacterial lineages [3]. Extensive experimental studies conducted over several decades, coupled with comparative analysis of several thousands of complete tailed phage genomes currently available in

IMG/M database at JGI using the accession numbers listed in S2 Data.

**Funding:** This work was supported by l'Agence Nationale de la Recherche grant ANR-20-CE20-0009-02 (to M.K.) and the European Union's Horizon 2020 research and innovation program under grant agreement 685778, project VIRUS X (to D.P.). Y.L. is a recipient of the Pasteur-Roux-Cantarini Fellowship from Institut Pasteur. The Ella and Georg Ehrnrooth Foundation and the Finnish Cultural Foundation are sincerely acknowledged (grants to T.D.). The work conducted by the U.S. Department of Energy Joint Genome Institute (S. R.) is supported by the Office of Science of the U.S. Department of Energy under contract no. DE-AC02-05CH11231. The development and application of GRAViTy analysis was supported by a grant to PS from the Wellcome Trust (WT108418AIA). The funders had no role in study design, data collection and analysis, decision to publish, or preparation of the manuscript.

**Competing interests:** The authors have declared that no competing interests exist.

**Abbreviations:** arCOG, archaeal clusters of orthologous genes; arTV, archaeal tailed virus; CGJ, composite generalized Jaccard; CP, carbamoylphosphate; dsDNA, double-stranded DNA; EC, Executive Committee; EOP, efficiency of plating; HMM, hidden Markov model; ICTV, International Committee on Taxonomy of Viruses; IMG/M, Integrated Microbial Genomes and Microbiomes; MCP, major capsid protein; MGM, modified growth medium; MIDAS, metal ion–dependent adhesion site; MTase, methyltransferase; ORF, open reading frame; REase, restriction endonuclease; RM, restriction–modification; RnR, ribonucleotide diphosphate reductase; SW, salt water; TA, toxin–antitoxin; vWA, von Willebrand factor type A.

the public sequence databases, have resulted in detailed understanding of the mechanisms that govern the biology, ecology, and evolution of this group of viruses [2,4–6]. Due to their ubiquity, tailed bacteriophages have a profound impact on the functioning of the biosphere through regulating the structure, composition, and dynamics of bacterial populations in diverse environments, from marine ecosystems to the human gut, and modulate major biogeochemical cycles [7–9]. Archaeal tailed viruses (arTV) are morphologically indistinguishable from tailed bacteriophages [1,4,10–12]. Similar to their bacterial virus relatives, the helical tails of arTVs can be short (podovirus morphology), long noncontractile (siphovirus morphology), or contractile (myovirus morphology) [13–15].

The arTVs have been thus far isolated on halophilic (class Halobacteria) and methanogenic (family Methanobacteriaceae) archaea, both belonging to the phylum Euryarchaeota [16–25]. Related proviruses have also been sighted in other lineages of the Euryarchaeota as well as in ammonia-oxidizing Thaumarchaeota and Aigarchaeota [26–30], whereas recent metagenomics studies revealed novel groups of arTVs putatively infecting marine group II Euryarchaeota, Thaumarchaeota, and Thermoplasmata [31–35]. Ecologically, it has been shown that virus-mediated lysis of archaea in the deep ocean is more rapid than that of bacteria, suggesting an important ecological role of archaeal viruses in marine ecosystems [36]. Evolutionarily, the broad distribution of tailed (pro)viruses in both bacteria and archaea suggests that viruses of this type were part of the virome associated with the last universal cellular ancestor, LUCA [3].

Genomic and structural analyses have shown that archaeal and bacterial tailed viruses have similar genomic organizations, with genes clustered into functional modules, and share homologous virion morphogenesis modules, including the major capsid protein (MCP) with the characteristic HK97 fold and the genome packaging terminase complex, suggesting common principles of virion assembly [11,13,22,30,37,38]. Nevertheless, at the sequence level, arTVs are strikingly diverse showing little similarity to each other and virtually no recognizable similarity to their bacterial virus relatives, indicative of scarce sampling of the archaeal virosphere [38]. Indeed, for several thousands of complete genome sequences of tailed bacteriophages [2], only 25 arTV isolates have been sequenced thus far. The low number of isolates severely limits our appreciation of the ecological impacts of arTVs and obscures the evolutionary history of this important and ancient group of viruses.

Virus discovery in the "age of metagenomics" is increasingly performed by culture-independent methods, whereby viral genomes are sequenced directly from the environment. This is a powerful approach that has already yielded thousands of viral genomes, providing unprecedented insights into virus diversity, environmental distribution, and evolution [39–46]. The limitation of viral metagenomics, however, is that the exact host species for the sequenced viruses typically remain unknown, and many molecular aspects of virus–host interactions cannot be accurately predicted. Here, to further explore the biology and diversity of arTVs, we sequenced the genomes of 37 viruses that infect different species of halophilic archaea and were isolated from hypersaline environments using classical approaches [16,17]. Collectively, our results provide the first global overview of arTV diversity and evolution and establish a taxonomic framework for their classification.

## Results and discussion

### Overview of new haloarchaeal tailed viruses

We sequenced a collection of 37 arTVs (5 siphoviruses and 32 myoviruses) infecting haloarchaeal species belonging to the genera *Halorubrum* and *Haloarcula* [16,17], more than doubling the number of complete genomes of arTVs (S1 Table). The sequenced viruses originate from

geographically remote locations, including Thailand, Israel, Italy, and Slovenia, and in combination with the previously described isolates, provide a substantially improved genomic insight into the global distribution of arTVs. The viruses possess double-stranded DNA (dsDNA) genomes ranging from 35.3 to 104.7 kbp in length. Genomes of several isolates were nearly identical (<17 nucleotide polymorphisms; S1 Table) and analysis of these genomes, in combination with host range experiments, was particularly illuminating toward the host range evolution among halophilic arTVs (see below).

## Archaeal tailed viruses represent a distinct group within the prokaryotic virosphere

To assess the global diversity of arTVs and analyze their relationship to bacterial virus members of the class *Caudoviricetes*, we supplemented the 37 genomes sequenced herein with the genomes of prokaryotic viruses available in GenBank and analyzed the dataset using GRAViTy [47,48] and vConTACT v2.0 [49]. The combined dataset included 63 complete arTV genomes (S1 Table), 3 of which were from viruses infecting methanogenic archaea, whereas all others were from haloarchaeal viruses. The GRAViTy tool classifies viruses into family-level taxonomic groupings according to homology between viral genes and similarities in genome organizations, which are expressed using composite generalized Jaccard (CGJ) distances [47,48]. We used a CGJ distance of over 0.8 as the threshold for family-level assignment, consistent with the family-level classification for eukaryotic viruses and the recently created families of bacterial viruses [47,48,50]. GRAViTy analysis of the global prokaryotic virome classified arTVs into 2 large assemblages, which could be further subdivided into 14 family-level groupings (CGJ distance ≥ 0.8) (Fig 1). To reveal a finer taxonomic structure within the arTV assemblage, we relied on the network analytics implemented in vConTACT v2.0, which has been specifically developed and calibrated to identify genus-level groupings of prokaryotic viruses [49]. Consistent with the GRAViTy results, the network analysis revealed 2 assemblages of arTVs, which were disconnected from all known bacterial and nontailed archaeal viruses (Fig 2). Notably, only a few genes were shared between arTVs and nontailed archaeal viruses, including those encoding a methyltransferase (MTase), transposase, adenylyltransferase, AAA+ domain protein, and proteins of unknown function (see spreadsheet "Fig 2 PCs" in S2 Data). Viruses within the 2 clades formed 23 genus-level and 14 family-level groups, many containing just 1 or 2 members, indicating that genetic diversity of archaeal viruses remains largely undersampled. The family-level and genus-level groupings are described in S1 Text.

Clade I in the GRAViTy analysis forms a sister branch to several groups of bacterial myoviruses, including families *Ackermannviridae*, *Herelleviridae*, and *Straboviridae* (T4-like bacteriophages), and consists of 4 family-level groups (Fig 1A and 1B, S1 Text), with family (F) 1 being by far the largest, with 39 members, which can be further divided into 4 genus-level (G) subgroups (Fig 2). Clade I is a cohesive assemblage consisting exclusively of haloarchaeal viruses and held together by 33 protein clusters, including those involved in virion morphogenesis, genome replication and repair, nucleotide metabolism, and several proteins of unknown function (S2 Table). By contrast, Clade II is less cohesive and comprises 10 family-level groupings, half of which consist of singletons (Fig 1A and 1C), and includes viruses of both halophilic and methanogenic archaea. Viruses from different family-level groups in this clade share an overlapping set of genes (52 protein clusters), including those involved in virion morphogenesis, genome replication, and repair, DNA metabolism and several other functionally diverse proteins. However, unlike in Cluster I, the set of genes holding the Cluster II together displays sporadic distribution, typically connecting 2 to 3 virus families (S2 Table).

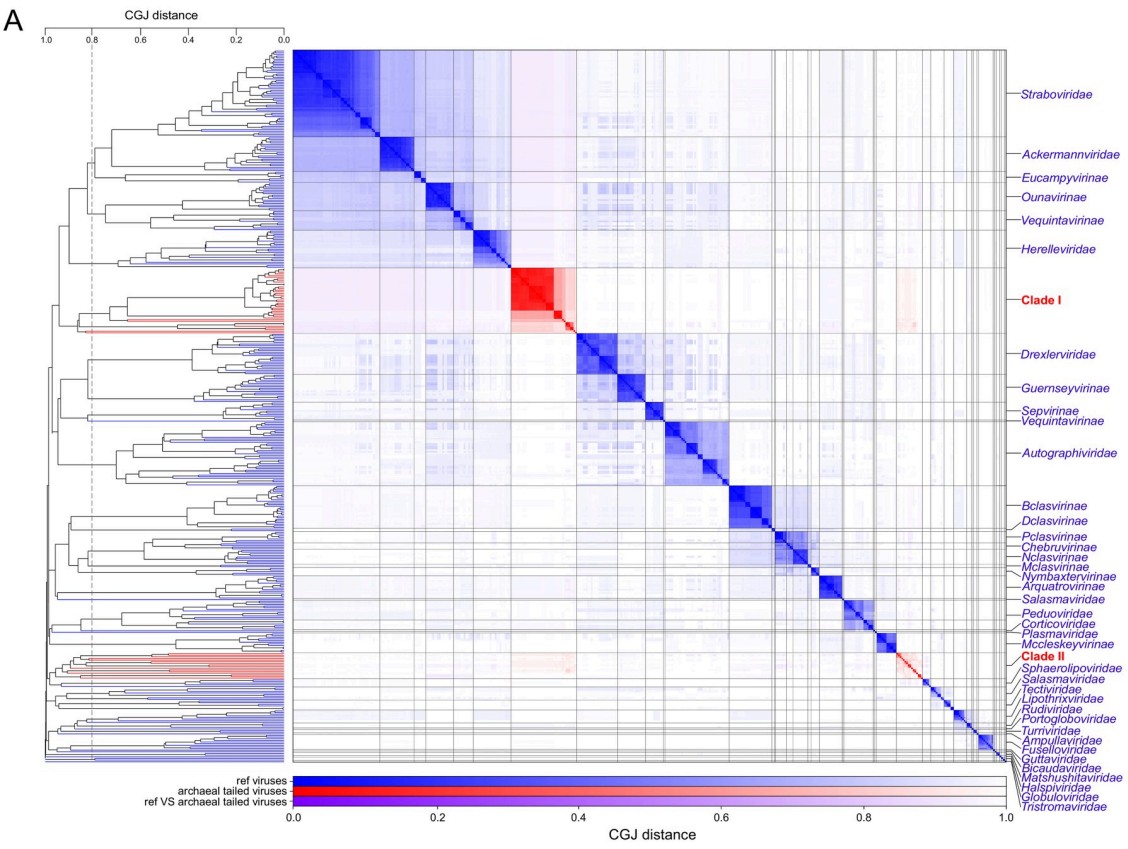

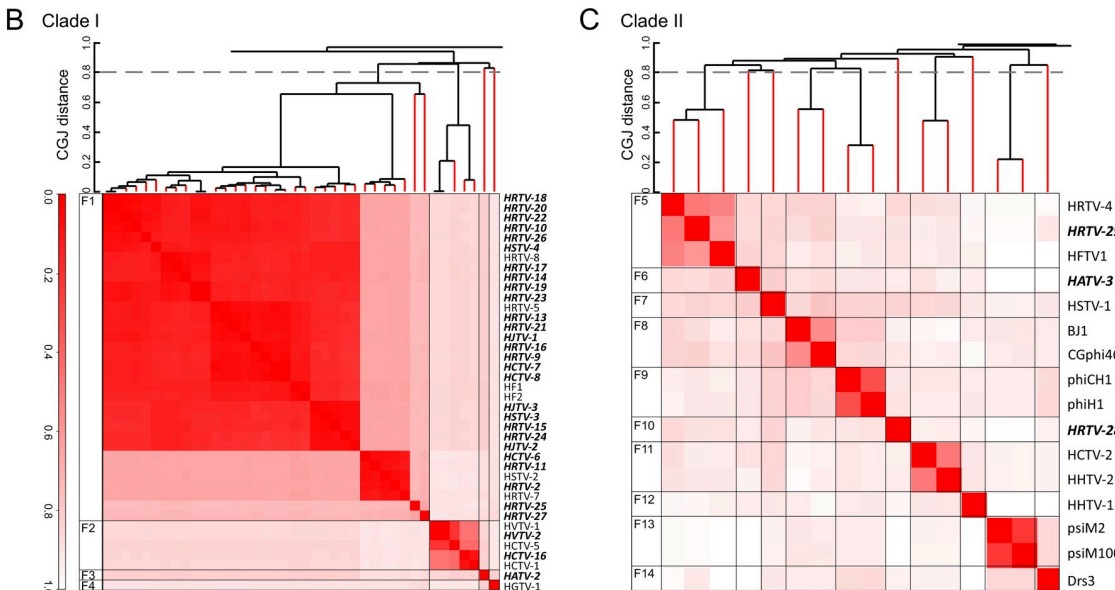

**Fig 1. Genome relationships between prokaryotic dsDNA viruses. (A)** Heat map and dendrogram of CGJ distances for classified bacteriophages and archaeal dsDNA viruses. Branches and clusters corresponding to arTVs are shown in red, whereas those of other viruses are in blue. **(B)** Zoom in on the arTV Clade I. **(C)** Zoom in on the arTV Clade II. Viruses sequenced in this study are highlighted in bold. CGJ distance of 0.8, chosen as a family-level threshold, is indicated with a broken line, with the family-level groups (F1 to F14) indicated on the left of the heat map. See S1 Data for the underlying dendrogram. arTV, archaeal tailed virus; CGJ, composite generalized Jaccard; dsDNA, double-stranded DNA.

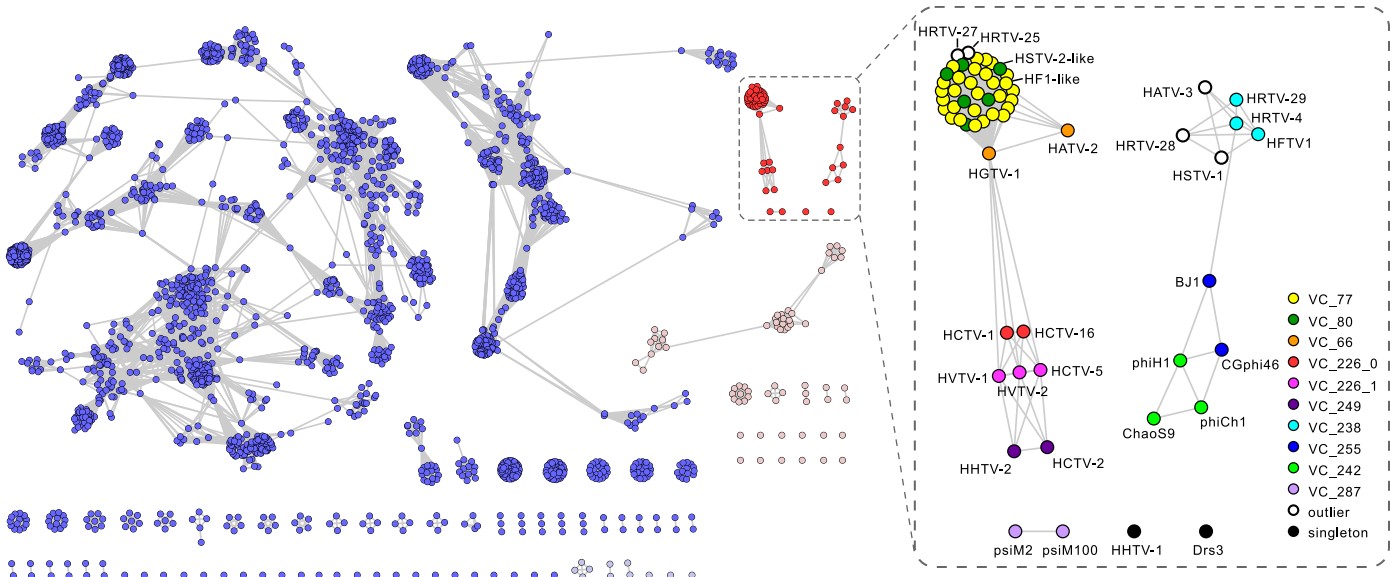

**Fig 2. The network-based analysis of PCs shared among arTVs and the prokaryotic dsDNA viruses.** The nodes represent viral genomes, and the edges represent the strength of connectivity between each genome based on shared PCs. Nodes representing genomes of arTVs are in red and other archaeal viruses are in pink, whereas those representing genomes of tailed bacteriophages are in blue and other bacteriophages are in light blue (left panel). The VCs of arTVs are enlarged and labeled in the right panel. See S2 Data for complete VCs and PCs generated by vConTACT v2.0. arTV, archaeal tailed virus; dsDNA, double-stranded DNA; PC, protein cluster; VC, viral cluster.

Based on the results of comprehensive comparative genomics analysis (see S1 Text), we propose classifying all known arTVs into 14 new families. The families *Hafunaviridae*, *Soleiviridae*, *Halomagnusviridae*, and *Pyrstoviridae* include viruses with icosahedral heads and long contractile tails (myovirus morphotype), whereas the families *Druskaviridae*, *Haloferuviridae*, *Graaviviridae*, *Vertoviridae*, *Suolaviridae*, *Saparoviridae*, *Madisaviridae*, *Leisingerviridae*, and *Anaerodiviridae* contain viruses with icosahedral heads and long noncontractile tails (siphovirus morphotype). The *Shortaselviridae* is the only family of viruses with short tails (podovirus morphology). The names of the 23 proposed genera are listed in S1 Table. Members of the same proposed genus typically share more than 60% of their proteins, and members of the same family share 20% to 50% of homologous proteins, whereas viruses from different families share less than 10% of proteins (Fig 3, S2 Fig). The 14 families have been approved by the Executive Committee (EC) of the International Committee on Taxonomy of Viruses (ICTV) and await ratification by the whole ICTV membership. Furthermore, the ICTV EC has approved unification of the 4 Clade I families into a new order *Thumleimavirales*; families *Haloferuviridae*, *Pyrstoviridae*, *Shortaselviridae*, and *Graaviviridae* have been included into a new order *Kirjokansivirales*; and families *Leisingerviridae* and *Anaerodiviridae* were included into a new order *Methanobavirales*.

## Gene content of archaeal tailed viruses

All virus genes were functionally annotated using sensitive profile–profile hidden Markov model (HMM) comparisons using HHpred [51] (S3 Table). The virus-encoded proteins were further classified into functional categories based on their affiliation to the archaeal clusters of orthologous genes (arCOG) [52] (S3 Table). Apart from the "Virus-related" proteins, the most frequent functional category assigned to archaeal virus proteins was the "Information storage and processing," with the "Genome replication, recombination, and repair" subcategory being

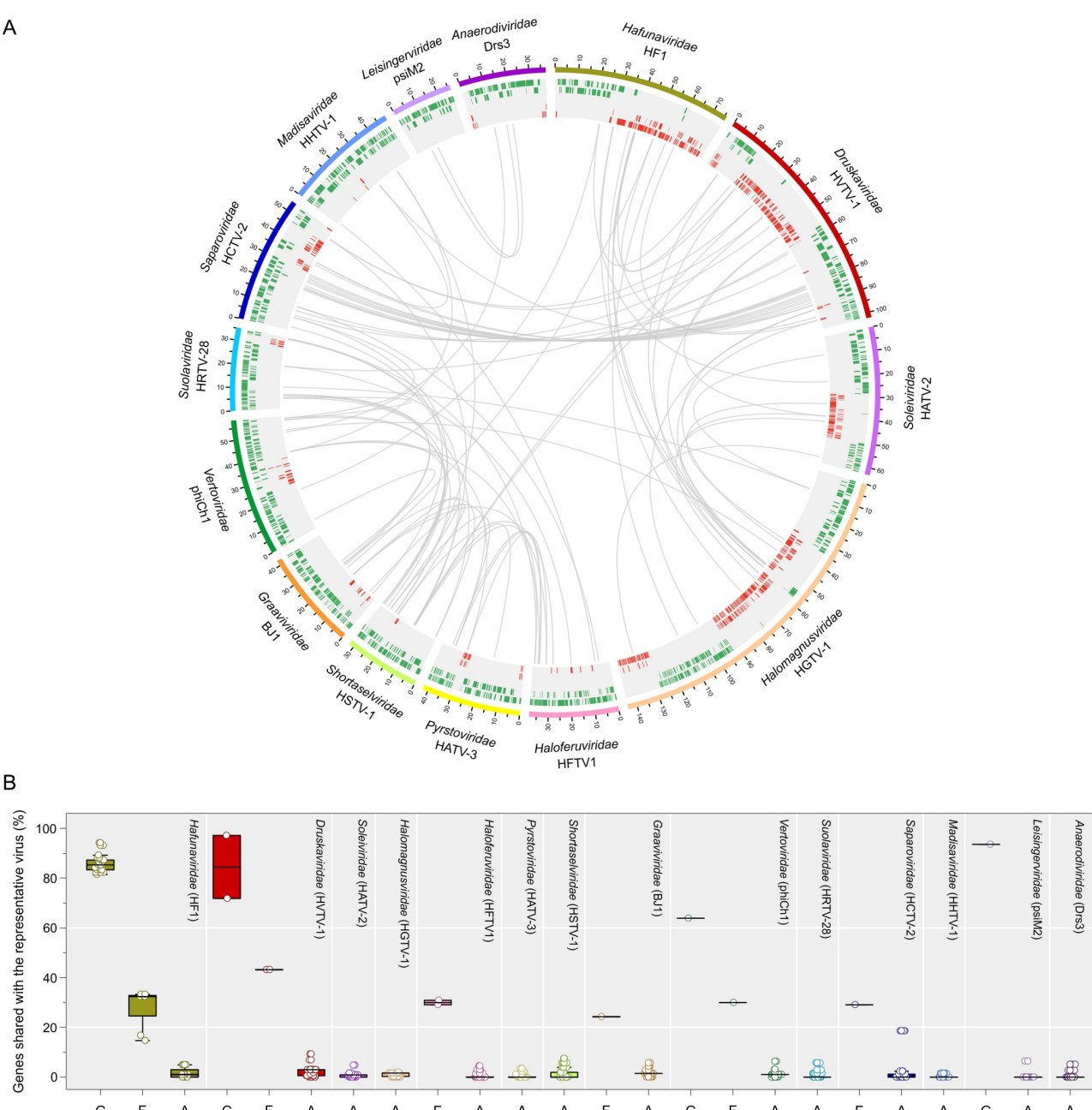

**Fig 3. Overview of homologous proteins shared by arTVs.** **(A)** The circus plot displays gene content similarity between representative members of each proposed family. Viral genomes from different families are indicated with distinct colors. The genomic coordinates (kbp) are indicated on genome maps. Putative ORFs distributed on 5 tracks (to avoid overlapping) are represented by green and red tiles on the forward and reverse strands, respectively. Homologous proteins are linked by gray lines. The representative viruses and the proposed virus family names are indicated. **(B)** The box plot shows the percentage of genes shared by a representative virus from each family with other members of the same genus (G) and family (F) as well as with arTVs from other families (A). Each box represents the middle 50th percentile of the data set and is derived using the lower and upper quartile values. The median value is displayed by a horizontal line. Whiskers represent the maximum and minimum values with the range of 1.5 IQR. Each virus is represented by a dot. In the circus and box plots analyses, proteins with over 30% amino acid sequence identity and E-value $< 1 \times 10^{-25}$ in our arTV database are considered to be homologous. Data underlying this figure can be found in S2 Data. arTV, archaeal tailed virus.

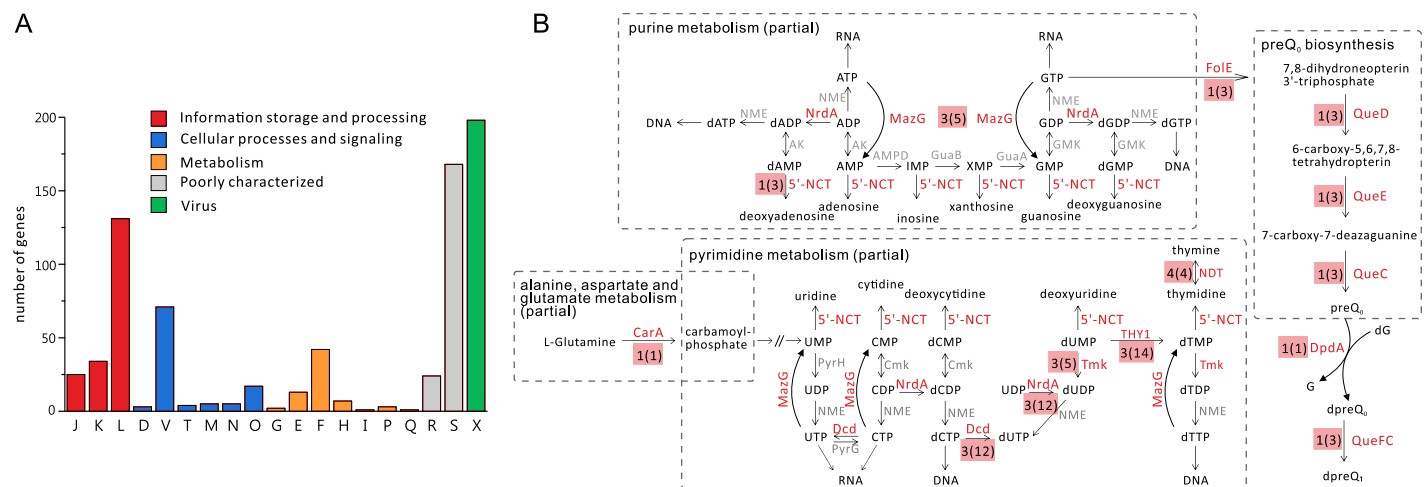

**Fig 4. Classification of genes encoded by arTVs. (A)** Classification of arTV genes into arCOG functional categories. The homologous gene shared by viruses in the same species is counted as one. The letters represent the following: J, translation, ribosomal structure and biogenesis; K, transcription; L, replication, recombination and repair; D, cell cycle control, cell division, chromosome partitioning; V, defense mechanisms; T, signal transduction mechanisms; M, cell wall/membrane/envelope biogenesis; N, cell motility; O, posttranslational modification, protein turnover, chaperons; G, carbohydrate transport and metabolism; E, amino acid transport and metabolism; F, nucleotide transport and metabolism; H, coenzyme transport and metabolism; I, lipid transport and metabolism; P, inorganic ion transport and metabolism; Q, secondary metabolites biosynthesis, transport and catabolism; R, general function prediction only; S, function unknown; X, virus related. **(B)** Schematic showing nucleotide-related (partial) metabolic pathways, with a particular enzyme indicated for each reaction. Enzymes that are found encoded by arTVs are in red, otherwise in gray. The number of virus families that harbor the enzyme is indicated next to the enzyme name, with the number of virus species shown in parenthesis. NrdA: RnR, MazG: NTP pyrophosphatase, 5′-NCT: 5′-deoxyribonucleotidase, CarA: CP synthase small subunit, Dcd: dCTP deaminase, Tmk: thymidylate kinase, THY1: thymidylate synthase thyX, NDT: nucleoside deoxyribosyltransferase, FolE: GTP cyclohydrolase I, QueC: queuosine biosynthesis protein, QueD: 6-pyruvoyl tetrahydropterin synthase, QueE: 7-carboxy-7-deazaguanine synthase, DpdA: paralog of queuine tRNA-ribosyltransferase, QueFC: NADPH-dependent 7-cyano-7deazaguanine reductase. See S3 and S5 Tables for individual genes classification. arCOG, archaeal clusters of orthologous gene; arTV, archaeal tailed virus; CP, carbamoylphosphate; RnR, ribonucleotide diphosphate reductase.

most strongly enriched (Fig 4A). Proteins of the "Defense mechanisms" subcategory from the "Cellular processes and signaling" category, primarily including diverse nucleases and DNA MTases, were also abundant. Finally, a substantial fraction of proteins was assigned to the "Metabolism" category, with proteins involved in nucleotide transport and metabolism being most common (Fig 4A). Below we highlight some of the observations with the more detailed description provided in S1 Text.

## Defense and counterdefense factors

Similar to tailed bacteriophages, arTVs are under strong pressure to overcome the host defenses, among which CRISPR–Cas, restriction–modification (RM) and toxin–antitoxin (TA) are the ubiquitous defense systems in both archaea and bacteria. No clear homologs of viral anti-CRISPR proteins were detected in the arTVs genomes, although several members of the *Hafunaviridae* and *Druskaviridae* encode a Cas-4–like nuclease that could potentially function in counterdefense against CRISPR–Cas systems (see S1 Text). By contrast, haloarchaeal tailed viruses ($n$ = 48) from 10 families encode diverse MTases with specificities for N6-adenine, N4-cytosine, and C5-cytosine (S4 Table, S3 Fig). The presence of the *MTase* genes can be linked to the presence of sequence motifs in the genomes of the corresponding viruses. For instance, GATC motif methylated by the phiCh1-like Dam MTase ($G^{m6}ATC$) [53] is present at high frequency (6.47 to 11.04/kbp) in the genomes of viruses that encode this MTase. By contrast, viruses that do not encode the Dam MTase also lack the corresponding motif in their genomes. Thus, arTVs have likely evolved to escape the host RM systems by either self-methylation via different virus-encoded MTases or purging the recognition motifs from their

genomes. In addition to the stand-alone MTases, some arTVs encode accompanying restriction endonucleases (REases), forming apparently functional RM systems, which could be deployed by the viruses for degradation of the host chromosomes or genomes of competing mobile genetic elements (see S1 Text). A similar function could be envisioned for the RNA- or DNA-specific micrococcal nucleases encoded by HRTV-17, phiH1, HCTV-2, and HHTV-2 (S3 Table).

The 7-deazaguanine modifications, produced by the virus-encoded preQ0/G+ pathway, were recently detected in the genomes of diverse viruses, including HVTV-1, and shown to confer viral DNA with resistance to various type II REases [54]. The preQ0/G+ pathway is encoded by all viruses of the *Druskaviridae* (Fig 4B, S3 Table), suggesting that these viruses depend on a similar genome modification for evasion of the host RM systems. Notably, HCTV-1 and HCTV-16 in addition encode a transporter of the queuosine precursor YhhQ. Interestingly, HRTV-29 (family *Haloferuviridae*) does not carry genes for the preQ0/G+ pathway, but encodes a DpdA homolog, a key enzyme mediating replacement of the unmodified guanine base in the DNA (Fig 4B, S3 Table). DNA modification with solely virus-encoded DpdA was demonstrated for the *Salmonella* phage 7–11 [54], suggesting that HRTV-29 also hijacks the host preQ0 pathway for DNA modification through its DpdA protein.

The TA systems are widely distributed in prokaryotes and have been shown to function in bacterial antiviral defense by initiating the programmed cell death, thereby preventing the virus spread [55]. Similar to certain marine bacteriophages [56], viruses of the families *Druskaviridae*, *Vertoviridae*, and *Madisaviridae* encode homologs of the nucleoside pyrophosphohydrolase MazG (S5 Table), which prevents the programmed cell death by degrading the intracellular ppGpp [57]. In addition, several arTVs encode homologs of the VapB antitoxins (arCOG08550), but not the associated toxins, suggesting a function in blocking the VapBC TA systems. Thus, arTVs appear to encode different factors for counteracting TA systems.

## Viral metabolic genes

The pangenome of arTVs includes many predicted metabolic genes with specific or general functions (S5 Table), such as phosphoadenosine phosphosulphate reductase (HRTV-29 and HFTV1), which could be involved in sulfur assimilation pathway; Class II glutamine amidotransferase (phiCh1 and ChaoS9); MaoC-like dehydratase domain protein (HATV-2), which exhibits (R)-specific enoyl-CoA hydratase activity [58]; dual specificity protein phosphatase (HGTV-1 and viruses from the genus *Mincapvirus* of the *Hafunaviridae*), thioredoxin (HGTV-1), peroxide stress response protein YaaA (HRTV-29 and ChaoS9), ADP-ribosyltransferase (HVTV-1 and HCTV-5), sialidase (HCTV-5) and Hsp90 chaperone protein (HSTV-2 and HGTV-1). However, the most common and widespread metabolic genes in arTVs appear to be linked to the replication, transcription, and translation of the viral genomes.

**Nucleotide biosynthesis.** Most haloarchaeal virus genes involved in DNA metabolism belong to the pyrimidine biosynthesis pathway and encode thymidylate synthase, thymidylate kinase, dCTP deaminase, and nucleoside deoxyribosyltransferase (Fig 4B). These proteins are broadly encoded by viruses from families *Hafunaviridae*, *Druskaviridae*, *Soleiviridae*, *Halomagnusviridae*, *Haloferuviridae*, and *Vertoviridae* (S5 Table). Interestingly, HGTV-1 (family *Halomagnusviridae*) encodes the small subunit, CarA, of carbamoylphosphate (CP) synthetase, which hydrolyzes glutamine to CP, a precursor common for the biosynthesis of pyrimidines and arginine [59] (Fig 4B). Thus, HGTV-1 might be redirecting the host metabolism toward de novo synthesis of pyrimidines. To our knowledge, no other viral isolate thus far has been reported to encode CarA/CarB subunits, although other enzymes in the de novo pyrimidine

synthesis pathway were found encoded by bacterial viruses [60]. Overall, the different genes identified suggest that haloarchaeal tailed viruses likely boost up the pyrimidine metabolism during infection. A similar behavior has been observed in *Pseudomonas* virus infections [61], suggesting that enhanced pyrimidine metabolism is critical for efficient replication of both archaeal and bacterial tailed viruses.

Viruses of the *Hafunaviridae*, *Druskaviridae*, and *Halomagnusviridae* encode class II ribonucleotide diphosphate reductases (RnRs), which convert rNTPs into corresponding dNTPs, the essential building blocks for the synthesis of viral DNA (Fig 4B, S5 Table). Viruses from the *Druskaviridae* encode putative 5′(3′)-deoxyribonucleotidases, which catalyze the dephosphorylation of nucleoside monophosphates, likely to regulate and maintain the homeostasis of nucleotide and nucleoside pools in the host cells [62] (Fig 4B, S5 Table).

Cobalamin (vitamin $B_{12}$) is an important cofactor in various metabolic pathways, including DNA biosynthesis (e.g., for class II RnR) [63]. Viruses of the *Druskaviridae* and *Saparoviridae* encode putative cobaltochelatase subunits CobS and CobT [EC Number: 6.6.1.2], which show sequence similarity to the 2 subunits encoded by certain tailed bacteriophages and cellular organisms (S5 Table). CobS is an AAA+ ATPase, whereas CobT contains a von Willebrand factor type A (vWA) domain and a metal ion–dependent adhesion site (MIDAS) domain located in the carboxyl-terminal region [64]. In HHTV-2, *cobT* is split into 2 open reading frames (ORFs) with the vWA domain encoded by a separate ORF. In prokaryotes, CobS interacts with CobT and, together with the third subunit, CobN, forms an active cobaltochelatase complex [65]. The latter catalyzes the insertion of $Co^{2+}$ ion into the corrin ring of hydrogennobyrinate a,c-diamide, close to the final step in the aerobic cobalamin biosynthesis pathway [63]. We found that the *cobST* 2-gene cluster is widely encoded in tailed viruses that infect members of 8 bacterial or archaeal orders, including Halobacteriales, as described in this study (S6 Table). In cyanophages, *cobST* gene cluster is part of the core genome [66], although *cobT* is usually mistakenly annotated as a peptidase. Recent bioinformatic study on the genotypes of *cob* subunits in prokaryotic genomes revealed that *cobS* and *cobT* are absent in most prokaryotic genomes, whereas *cobN* is present [65]. Many such prokaryotes, including haloarchaea, were suggested to employ a mosaic cobaltochelatase consisting of CobN and magnesium chelatase subunits [65,67]. Our finding of the broad distribution of the *cobST* gene cluster in tailed viruses, both bacterial and archaeal, suggests that viral CobST likely hijacks the host CobN (from the mosaic cobaltochelatase in the absence of the host encoded CobST), to increase the production of cobalamin, thereby promoting the virus replication.

**Transcription and translation.** It has been previously shown that arTVs encode many proteins involved in RNA metabolism [38], and our current study further expands the complement of viral genes implicated in RNA metabolism and protein translation. For instance, HJTV-2 (*Hafunaviridae*) encodes a Trm112 family protein, which in *Haloferax volcanii* interacts and activates 2 MTases, Trm9 and Mtq2, known to methylate tRNAs and release factors, respectively [68,69]. Remarkably, 4 viruses, HRTV-24, HSTV-3, HJTV-3, and HRTV-15 (*Hafunaviridae*), encode homologs of the ribosomal protein L21e (S5 Fig), which mediates the attachment of the 5S rRNA onto the large ribosomal subunit and stabilizes the orientation of adjacent RNA domains [70]. Ribosomal proteins have been recently shown to be encoded by diverse tailed bacteriophages and at least some of these proteins can be incorporated into host ribosomes [71]. The L21e homologs identified herein are the first examples of ribosomal proteins encoded by archaeal viruses.

*Halomagnusvirus* HGTV-1 encodes a record number of tRNA genes among archaeal viruses: 36 tRNA genes corresponding to all 20 proteinogenic amino acids [38]. In addition, this virus has a number of tRNA metabolism genes: 2 Rnl2-family RNA ligases, which may be involved in the removal of tRNA introns [38]; a tRNA nucleotidyl transferase (CCA-adding

enzyme) involved in tRNA maturation; a Class I lysyl-tRNA synthetase which catalyzes the attachment of lysine to its cognate tRNA; and a tRNA splicing ligase RtcB possibly responsible for the repair of viral tRNAs. RtcB and several tRNAs are also encoded by viruses from the *Hafunaviridae* and *Druskaviridae* (S1 Table).

**Integrases.** All members of *Hafunaviridae*, *Graaviviridae*, *Vertoviridae* and *Leisingerviridae* as well as HCTV-5 from *Druskaviridae* (but not other members of this family) encode site-specific tyrosine recombinases. Consistently, analysis of the archaeal genome sequences available in NCBI database revealed the presence of apparently functional (based on the conservation of all key proteins), integrases-encoding proviruses from the former 4 families, including several previously reported proviruses [20,22] (S7 Table, S6 Fig), suggesting a temperate life cycle for members of the *Hafunaviridae*, *Graaviviridae*, *Vertoviridae* and *Leisingerviridae*. All proviruses were flanked by recognizable direct repeats corresponding to the attachment sites (S7 Table). By contrast, the recombinase of HCTV-5 is homologous to the invertase encoded by viruses of *Vertoviridae*, which is responsible for the invertion of the tail-fiber module of phiCh1 [72]. Thus, HCTV-5 recombinase is likely to be involved in genome rearrangements rather than site-specific integration. Consistently, proviruses related to HCTV-5 or other members of the *Druskaviridae* were not identified. Searches for proviruses related to other arTV families revealed that members of the families *Haloferuviridae* and *Anaerodiviridae* can also have temperate members (S7 Table, S6 Fig), although none of the current isolates encodes an integrase, suggesting that the integration module and hence the integration ability can be occasionally gained by arTVs. In total, 38 proviruses belonging to 6 arTV families (based on the same demarcation criteria which we used to define the families) were identified in diverse species of halophilic and methanogenic archaea (S7 Table), considerably expanding the genetic diversity and distribution of the corresponding virus families. The majority of haloarchaeal proviruses were integrated into diverse tRNA genes, whereas proviruses belonging to the *Anaerodiviridae* were integrated either into intergenic regions or protein-coding genes (S7 Table). No proviruses related to members of the other 8 families could be identified.

## Mutations in tail fiber genes determine the broad host range of hafunaviruses

The arTVs from different families display highly variable host ranges, with some being specific to a single isolate and others infecting multiple species from different haloarchaeal genera [16,17,22,25]. We did not observe obvious relationship between the host range and virus taxonomy, with some representatives of the same family displaying a broad host range and others being able to infect only 1 strain. Thus, host range determinants appear to be virus specific. Unlike in the case of tailed bacteriophages, the factors underlying the host range in haloarchaeal viruses remain obscure. To gain insights into this question, we focused on viruses of the family *Hafunaviridae* which, with 39 myovirus isolates, is currently the largest family of arTVs. Furthermore, most hafunaviruses have broad host ranges, collectively being able to infect a dozen of different strains belonging to 5 haloarchaeal genera [16,17,22]. We assessed the efficiency of plating (EOP) of 24 hafunaviruses (19 isolates from 5 species in the genus *Haloferacalesvirus* and 5 isolates from a single species in the genus *Mincapvirus*) on a panel of 24 haloarchaeal strains from 5 genera [16,17]. The host ranges of viruses within each species varied considerably (S8 Table), consistent with the previous results [16,17].

Availability of complete genome sequences for multiple isolates from the same species allowed us to pinpoint the genes responsible for the observed differences in the host range. The most informative was the comparison of 2 isolates, HRTV-19 and HRTV-23, which differ by 2 nucleotide substitutions, but their EOPs on 2 of the tested strains differ by 3 orders of

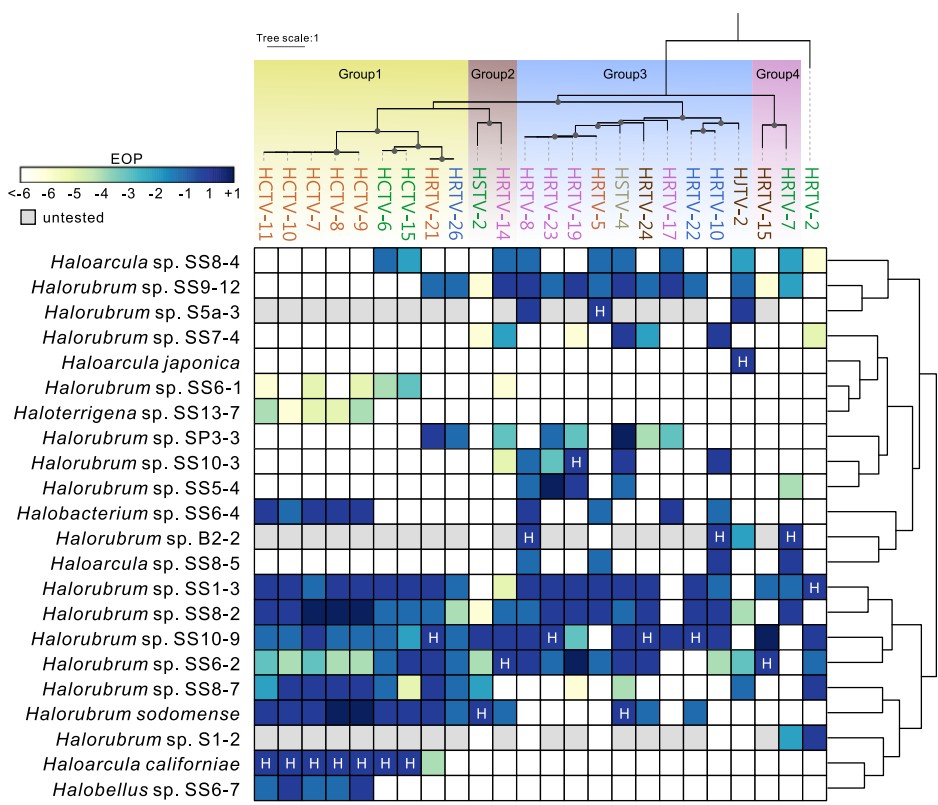

**Fig 5. Host ranges of hafunaviruses.** The heat map shows the EOP of hafunavirus isolates on different haloarchaeal strains. The EOP of the virus on its original isolation host has been set to 1 and marked with H. Number 1 is equal to the EOP on the original host. Numbers −1, −2,. . . refer to $10^{-1}$, $10^{-2}$,. . . and +1 refers to $10^1$ when compared to the EOP on the original host, as shown in S8 Table. The isolation host for HRTV-26 is *Halorubrum* sp. SS13-13, which occasionally did not form a lawn and was not included in this analysis. The titer of HRTV-26 on *Halorubrum* sp. SS13-13 was $2 \times 10^8$ plaque-forming units per milliliter, which was considered here as 1 and used for comparison. The hosts were clustered based on the similarity of EOPs of the tested viruses, as represented by the dendrogram on the right side. The upper panel shows the maximum likelihood phylogenetic tree of the tail fiber adhesin proteins encoded by the analyzed viruses. The adhesin of HRTV-2, the amino acid sequence of which is markedly divergent from those of other hafunaviruses, is set as the outgroup. The 4 distinct groups of adhesins are displayed with colored blocks. Bootstrap values greater than 90% are indicated in the nodes by dots. Virus names are colored according to the species to which they belong. EOP, efficiency of plating.

magnitude (Fig 5, S8 Table). The 2 mutations mapped to 2 adjacent genes located at the end of the tail morphogenesis module, suggesting that this locus plays a key role in determining the host specificity. This possibility is further supported by the observation that 5 isolates (HCTV-7, HCTV-8, HCTV-9, HCTV-10, and HCTV-11) in which the 2 genes are identical but sequence divergence in other loci ranges from 1 bp to approximately 1.2 kbp share the same host ranges, with similar EOPs (Fig 5, S8 Table). The first of the 2 genes encodes a glycine-rich tail fiber protein and the second gene encodes a small putative protein (S7 Fig). The glycine-rich protein displays features typical of adhesin proteins located at the distal tip of the tail fibers of diverse T-even phages and has been shown to determine the host specificity in phages [73]. The adhesins encoded by hafunaviruses form 4 distinct phylogenetic groups (Groups 1 to 4), with the corresponding viruses displaying distinct infectivity patterns on the tested haloarchaeal strains (Fig 5, S8 Fig). This is most prominent in the case of viruses encoding the largest adhesin groups, Group 1 and Group 3. For instance, 4 strains, *Halorubrum* sp. SS8-7, *Halorubrum sodomense*, *Haloarcula californiae*, and *Halobellus* sp. SS6-7, are particularly sensitive to

viruses encoding Group 1 adhesins, with the latter 2 strains being exclusively infected by viruses encoding this group of adhesins. Conversely, no virus of this group is able to infect *Halorubrum* strains SS7-4, SS10-3, and SS5-4 or *Haloarcula* sp. SS8-5. Nearly all viruses encoding Group 3 adhesins are able to infect *Halorubrum* sp. SS9-12, whereas *Halorubrum* sp. SS6-1 and *Haloterrigena* sp. S13-7 are completely insensitive to this group of viruses (Fig 5) (see S1 Text for details). In all hafunaviruses, the adhesin gene is located within a hypervariable region (S7 Fig), which has been previously suggested to encode tail fiber proteins [22]. Collectively, our results implicate the adhesin-encoding gene as the key host range determinant in hafunaviruses. Importantly, while the host range pattern can be explained, even if partly, by the adhesin phylogeny, overall intergenomic relationships between hafunaviruses is a poor predictor of the host range. Indeed, viruses belonging to the same species (i.e., >95% genome-wide identity) encode adhesins from different phylogenetic groups and display distinct host ranges.

## Continuity between archaeal tailed viruses across biomes

The previous study of viral communities using metagenomics approach revealed abundant presence of tailed viruses in hypersaline environments [74]. To assess the extent of sequence diversity and environmental distribution of halophilic arTVs and to explore the evolutionary relationships between tailed viruses from different environments, we searched the Integrated Microbial Genomes and Microbiomes (IMG/M) database for homologs of MCP, portal and the large terminase (TerL) proteins representing each family of arTVs. The searches collectively yielded 117,049 contigs encoding at least 1 of the 3 viral hallmark proteins. The dataset was supplemented with reference sequences from viruses for which hosts are known or predicted in silico, including tailed (pro)viruses associated with methanogens, MGI and MGII Euryarchaeota, Thaumarchaeota, Thermoplasmata, and Aigarchaeota [26,27,29,31–35]. In addition, sequences of uncultivated haloviruses previously obtained through fosmid sequencing from the Santa Pola saltern, Spain, were also added to the dataset [75]. Among the 3 hallmark proteins, TerL appeared to be the least specific to archaeal viruses, likely due to its relatively high conservation across all tailed bacterial and archaeal viruses. Accordingly, searches queried with haloarchaeal virus TerL against the IMG/M database retrieved a high number of significant hits from diverse environments (S11 Fig). By contrast, MCP and portal proteins were more specific to halophilic arTVs and retrieved a higher ratio of homologs from hypersaline ecosystems compared to those from other habitats. This is consistent with the fact that haloviral MCPs and portal proteins typically display low sequence similarity to their bacteriophage homologs in BLASTP searches. Phylogenetic analysis of the arTV MCP and portal proteins produced largely congruent trees (S12 Fig), suggesting that the 2 proteins coevolve and the corresponding genes rarely undergo nonorthologous replacement. The only possible exceptions to this trend include HGTV-1 and ChaoS9, with the latter being notoriously chimeric [20] (S1G Fig). Consequently, MCP and portal proteins were used as specific markers for arTVs.

The global MCP tree splits into 9 well-supported clades, each represented by haloarchaeal tailed virus isolates from one or more families (Fig 6). Structural comparison of the MCPs from representatives of the 14 arTV families as well as selected uncultured arTVs revealed the conservation of the HK97 fold (Fig 7A) and recapitulated the same 9 clades obtained using sequence-based phylogenetic analysis (Fig 7, S1 Text), testifying to the robustness of these groups. Finally, a similar clustering is also observed in the portal tree (S11 and S12 Figs). In the global MCP phylogeny including sequences from metagenomic databases, almost all clades in the MCP tree contain numerous homologs originating from hypersaline environments (Fig 6),

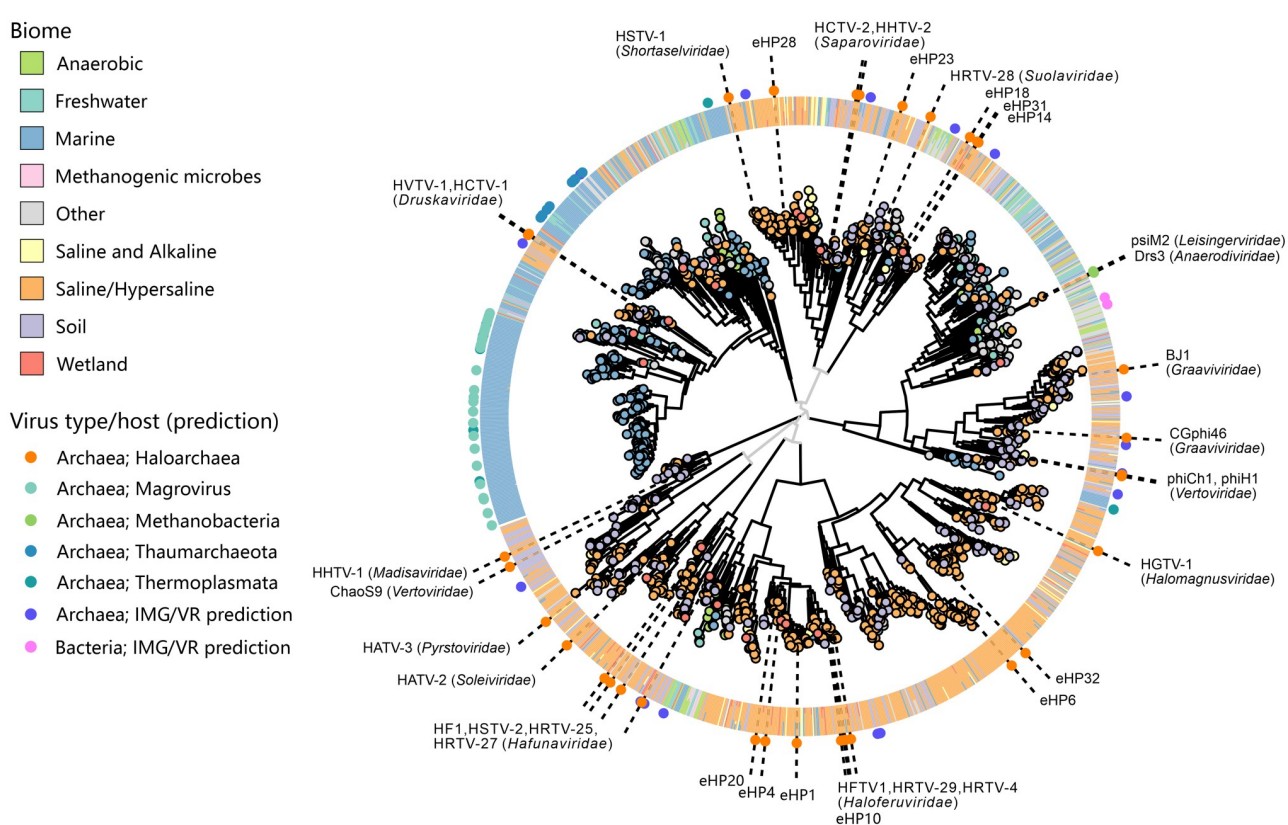

**Fig 6. Phylogenetic analysis of the MCPs encoded by arTVs.** The homologs retrieved from metagenomes were clustered at 0.9 protein sequence identity and the representative of each cluster is indicated as a circle at the tip of the branch with the color reflecting the corresponding biome (see the legend on the left). The ring outside of the tree consists of tiles indicating the biome compositions from the corresponding protein cluster. The dot in the outermost ring indicates the (predicted) virus type in the corresponding branch and is colored based on the host predicted. The names of the arTV isolates of haloarchaea and methanogens as well as the fosmids generated from a saltern of Santa Pola in Spain [75] are shown on the tree. The virus family names are indicated next to the corresponding arTV isolates. The associated data on metagenomic contigs from the IMG/M database can be found in S2 Data. arTV, archaeal tailed virus; IMG/M, Integrated Microbial Genomes and Microbiomes; MCP, major capsid protein.

with the corresponding metagenomes derived from all 7 continents, testifying to the wide geographic distribution of the halophilic arTVs. This result suggests that the haloarchaeal virus isolates described herein adequately represent the overall diversity of haloarchaeal tailed virus communities.

Interestingly, although searches with the hallmark proteins were performed against the global IMG/M database, only a handful of sequences affiliated to bacteria/phages were retrieved (Fig 6, S10 Fig). By contrast, haloarchaeal viruses formed well-supported clades with the previously characterized viruses associated with marine and methanogenic archaea, suggesting that all arTVs are more closely related to each other than to tailed bacteriophages, consistent with the GRAViTy analysis (Fig 1). Indeed, 3 of the clades in the MCP tree contained considerable fractions of taxa originating from nonhypersaline environments (Fig 6). The largest of these clades is dominated by MCP homologs from marine environments, with a relatively small subclade including haloarchaeal viruses of the *Druskaviridae* (e.g., HVTV-1), interspersed between reference viruses associated with MGII Euryarchaeota (magroviruses) [34,35], marine Thaumarchaeota [32], and Thermoplasmata [33] (Fig 6). A similar clustering

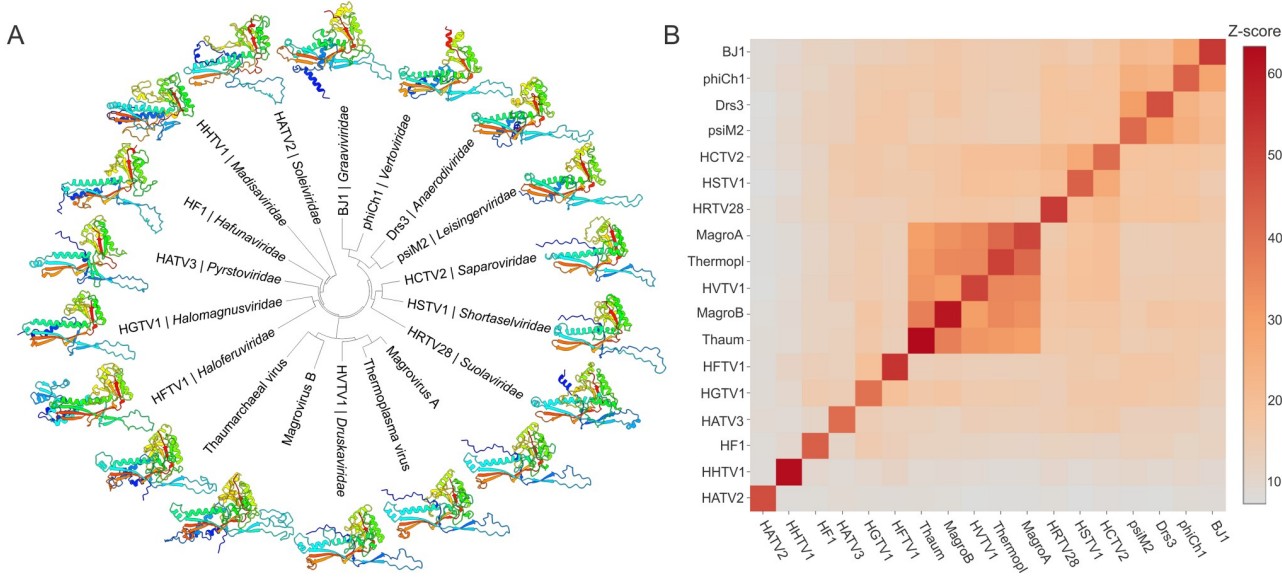

**Fig 7. Structural comparison of the arTV MCPs. (A)** Structural models of arTV MCPs shown using ribbon representation and colored using rainbow scheme from the N-terminus (blue) to C-terminus (red). The models are shown next to the corresponding tips of the cladogram based on the pairwise structural similarity (Z) scores. **(B)** All-against-all comparison of the arTV MCP structural models shown as a matrix of pairwise Z scores generated using DALI [93]. The underlying data can be found in S2 Data. arTV, archaeal tailed virus; MagroA/B, group A and B Magroviruses; MCP, major capsid protein; Thaum, a virus of marine Thaumarchaeota; Thermopl, a virus of marine Thermoplasmata.

is also retrieved in the portal protein tree (S11 and S12 Figs), suggesting that the ancestor of the *Druskaviridae* has evolved from the midst of marine archaeal viruses.

The second MCP tree clade containing sequences from diverse environments is represented by viruses from the *Graaviviridae* and *Vertoviridae* (e.g., BJ1 and phiCh1, respectively; note that the MCP of ChaoS9 from the *Vertoviridae* clusters with that of HHTV-1 from the *Madisaviridae*). This group of halovirus MCPs is embedded within an assemblage of homologs from diverse nonsaline environments, including anoxic habitats (Fig 6). Consistently, the clade includes viruses psiM2 and Drs3 (families *Leisingerviridae* and *Anaerodiviridae*, respectively) infecting methanogenic archaea (Fig 6). The same relationship is recapitulated in the portal protein tree (S11 and S12 Figs). Therefore, viruses from the families *Graaviviridae* and *Vertoviridae* appear to have evolved from viruses inhabiting moderate, nonsaline environments, although the exact evolutionary history remains obscure, given the complicated composition and relatively poor sampling of this clade.

Finally, the third MCP tree clade is represented by viruses from the families *Haloferuviridae* and *Halomagnusviridae* (e.g., HFTV1 and HGTV-1, respectively). Unlike in the other 2 cases, where haloarchaeal viruses were nested among taxa from nonhypersaline environments, in this clade, we observe a reverse situation. Namely, a group of sequences, largely originating from anoxic environments, such as methanogenic microbial communities, anaerobic digesters, wetlands, etc., with at least 1 contig predicted to represent a virus infecting a methanogenic archaeon, forms a sister clade to haloviruses of the *Haloferuviridae* and is further embedded within a clade dominated by sequences from hypersaline environments (Fig 6). A parsimonious explanation for such clustering involves evolution of a group of methanogenic archaeal viruses from within the diversity of haloarchaeal viruses.

Collectively, these results suggest that there is a continuity between arTVs across different biomes. At the same time, however, it becomes clear that haloarchaeal viruses represent a poly-phyletic assemblage, with several groups immigrating into hypersaline environments from other habitats with lower salinity. Presumably, the ancestors of these groups gained the ability to infect halophilic archaea by accumulating mutations within the tail fiber genes, akin to those described above for hafunaviruses. We note that natural mixing between marine and hypersaline ecosystems (e.g., through evaporation of coastal marine water) provides an eco-logical setting for marine viruses to encounter halophilic archaea. Thus, hypersaline environ-ments, once considered as an isolated extreme environment [76,77], emerge as an integral part of the whole ecosystem.

## Concluding remarks

In the present study, we sequenced 37 haloarchaeal tailed virus isolates, which originate from 4 spatially remote locations, more than doubling the current number of sequenced arTVs (S1 Table). Unlike for some hyperthermophilic archaeal viruses [78], for most arTVs, there is no relationship between genetic closeness (and hence taxonomy) and the site of virus isolation. For 6 out of the 7 virus families with more than 1 isolate (*Hafunaviridae*, *Druskaviridae*, *Halo-feruviridae*, *Graaviviridae*, *Vertoviridae*, and *Leisingerviridae*), members were isolated from 2 to 5 geographically remote locations (S1 Table). In the cases of families *Hafunaviridae* and *Druskaviridae*, which contain members belonging to the same species (i.e., >95% identical at the genome level), nearly identical viruses were isolated from remote sites, indicating global distribution of certain arTV species. Through a systematic analysis of all 63 available arTVs, we establish the first framework for the taxonomy of arTVs and propose 14 new families and 3 orders for their classification to be included within the class *Caudoviricetes* alongside tailed bacteriophages. Half of the arTV families are currently represented by a single virus isolate, indicating the scarce sampling of haloarchaeal virome. While this manuscript was under review, 3 new haloarchaeal viruses and one virus infecting methanogenic archaea have been described [22,79,80]. *Halorubrum* viruses Serpecor1 and Hardycor2 belong to the family *Hafu-naviridae* [22], whereas *Halorubrum* virus Hardycor1 [79] and *Methanocaldococcus* virus MFTV1 [80] are only distantly related to other arTVs and will likely represent new virus families.

It has been previously shown that arTVs share virion architecture, assembly principles, and genome organization with their bacterial virus counterparts [11,13,30,38,81]. In this study, we focused on metabolic genes and showed that arTVs carry a rich repertoire of counterdefense and metabolic genes, predominantly those involved in DNA and RNA metabolism. Many of these genes are also commonly encoded by tailed bacteriophages, suggesting common propa-gation strategies for these bacterial and archaeal viruses. Nevertheless, some of the identified metabolic genes have never been reported in other viruses, e.g., *carA*, which is predicted to promote the de novo synthesis of pyrimidines, RNA MTase activator *trm112* implicated in modulation of protein translation, and the gene encoding a putative ribosomal protein L21e.

Myoviruses of the *Hafunaviridae*, the most populous family of arTVs, have broad host ranges, with strains of the same species displaying considerable variation in the host range. We found that the host range specificity in these viruses is primarily determined by mutations within a gene encoding a tail adhesin protein, which resembles that of T-even bacteriophages [73,82]. High sequence divergence of adhesins encoded by hafunaviruses likely allows these viruses to explore a broad landscape of receptor molecules on the host surface, boosting the competitiveness of this group of viruses in the environments with some of the highest reported virus-to-host ratios. Despite the fundamental differences of archaeal and bacterial cell surface

structures [83], the host range determinant of tailed viruses, the tail adhesins, can be highly similar in sequence features between arTVs and tailed bacteriophages. This highlights the evolutionary potential of the tail fibers adapt to diverse host receptors, which may underlie the overall success of the *Caudoviricetes* viruses.

Finally, the survey of metagenomic databases uncovered a considerable diversity and global distribution of arTVs. Our results point to the polyphyletic origins of the haloarchaeal tailed viruses, with some of the groups originating from archaeal viruses associated with marine and methanogenic hosts, indicative of virus movement across different biomes. Given that in hypersaline environments the arriving viruses are exposed to extremely high salt concentrations both inside and outside of the cell [84], the adaptation must have entailed a substantial evolution of the virion proteins. Indeed, some of the halophilic arTVs have been shown to be noninfectious under low salt conditions [11,37]. However, the effect was reversible and upon addition of salt, the infectivity was restored, suggesting that virions of haloarchaeal viruses can maintain stability and integrity under a wide range of environmental conditions. Further sampling and isolation of arTVs, especially those infecting nonhalophilic archaea, will be important for further understanding the ecology and evolution of this important group of viruses.

## Materials and methods

### Preparation of virus samples

Viruses sequenced in this study had been isolated previously [16,17]. The host strains used for virus production are listed in S1 Table. Viruses and strains were grown aerobically at 37˚C in modified growth medium (MGM) made using artificial salt water (SW) (23% (w/v) in broth, 20% in solid, and 18% in top agar) (http://www.haloarchaea.com/resources/halohandbook/). The double-layer plaque assay method was used for virus growth and quantification, and virus stocks were prepared from semiconfluent plates as previously described [16,17]. Viruses were precipitated from stocks with polyethylene glycol 6000 (final concentration of 10% (w/v)) by magnetic stirring for 1 hour at 4˚C, collected by centrifugation (Fiberlite F14 rotor, 9,820 g, 40 minutes, 4˚C), and resuspended in 18% SW. Virus samples were purified by rate zonal centrifugation in 5% to 20% (w/v) sucrose (18% SW; Sorvall AH629 rotor, 103,586 g, 40 minutes, 15˚C), concentrated by differential centrifugation (Sorvall T647.5 rotor, 113,580 g, 3 hours, 15˚C) and resuspended in 18% SW.

### Viral genome isolation and sequencing

Nucleic acids were extracted from purified virus samples either using the PureLink Viral RNA/DNA Mini Kit (Thermo Fisher Scientific, Germany) or by phenol/ether method combined with ethanol precipitation. The viral genome libraries were prepared with the Nextflex PCRFree kit (Bioo Scientific, Austin, Texas) and sequenced by Illumina Miseq (Illumina, San Diego, California, USA) with paired-end 250-bp read length (Biomics Platform, Institut Pasteur, France). Sequenced reads were quality-trimmed using fqCleaner v0.5.0 and assembled using clc_assembler v4.4.2 implemented in Galaxy-Institut Pasteur. All genome sequences were submitted to GenBank, and the accession numbers are provided in S1 Table.

### Viral genome annotation and bioinformatics analysis

The virus genomic termini and packaging mechanism were determined by PhageTerm [85]. ORFs were predicted using RAST v2.0 server [86]. Functional annotation of viral proteins was performed using HHpred [51]. The average nucleotide identity between viruses was calculated by ANI Calculator [87], whereas the global alignment of amino acid sequences was carried out

by EMBOSS Needle tool [88]. Multiple sequence alignments were performed by PROMALS3D [89]. Phylogenetic analyses were carried out using PhyML 3.0, with the automatic model selection [90]. To identify the proviruses related to arTVs, genome sequences of representatives of each proposed genus were used as seeds for blastx searches against the nonredundant protein sequence database at NCBI with the E-value cutoff of $1 \times 10^{-5}$. The exact nucleotide coordinates and attachment sites were identified manually.

## Structural modeling and comparisons

Structural modeling for all but one arTV MCPs was performed using AlphaFold2, with the mmseq2 algorithm for collection and alignment of homologs [91]. In the case of HHTV-1 MCP, AlphaFold2 failed to produce a satisfactory model and hence RoseTTAFold was used instead [92]. The resulting structural models were manually inspected and, when present, the N-terminal extensions corresponding to the scaffolding domains were removed. All structural models were then compared to each other using DALI [93] to produce a matrix and a dendrogram of structural similarity.

## Taxonomy assignment

The viral genomes were subjected to GRAViTy [47,48] (http://gravity.cvr.gla.ac.uk) and vConTACT v2.0 [49] for taxonomy assignment. We incorporated the annotated genomes of arTVs into the genomic datasets that were (i) built for bacterial and archaeal dsDNA viruses from the ICTV 2016 Master Species List 31V1.1 for GRAViTy analysis; and (ii) embedded in vConTACT v2.0 with the NCBI Bacterial and Archaeal Viral RefSeq V88 and default settings for analysis. The virus network generated by vConTACT v2.0 was visualized with Cytoscape software v.3.7.2 (https://cytoscape.org/) using an edge-weighted spring-embedded model that genomes sharing more PCs were positioned closer to each other. Pair-wise genomic comparison was performed by EasyFig with the E-value cutoff of $1 \times 10^{-3}$ and minimum protein length of 15 residues [94]. The overall similarity among arTVs on genus and family levels were analyzed and proteins with over 30% amino acid sequence identity and E-value $< 1 \times 10^{-25}$ in our arTV database were counted as homologous. The results were visualized with Circos plot [95], whereas box plots were prepared using OriginLab v9.8 (OriginLab, USA).

## Virus–host interactions

Virus–host pairs tested for hafunaviruses in this study are shown in S8 Table. For initial screening of virus–host interactions, 10-μl drops of undiluted and 100-fold diluted virus stocks were spotted on the lawn of early-stationary growing strains prepared by the double-layer method, and 10-μl drops of MGM broth were spotted as negative control. After 2 to 5 days of incubation, plates were screened for the growth inhibition. When inhibition was observed, plaque assay was used to verify virus–host interactions and quantify the EOP. The titers above $10^3$ plaque-forming units per milliliter could be detected. Data were normalized by comparing the EOP to that on the original host strain (set as 1). The heat map of the virus host ranges was generated based on the EOP results and clustered according to the phylogeny of viral adhesins as well as EOP similarity of the tested hosts using pheatmap package in R.

## Metagenomic database screening

The MCP, portal, and TerL protein sequences of the isolated halophilic arTVs were used as queries in a BLAST search of 31,344 public metagenomes available in the IMG/M database at JGI with a threshold of 100 bit score and 80% sequence coverage. Sequences of the 3 markers

from available (pro)viruses of methanogen, marine group I and group II Euryarchaeota, Thaumarchaeota, Thermoplasmata, Aigarchaeota, and fosmids from a saltern of Santa Pola in Spain were extracted [26,27,29,31–35,72]. These sequences were subjected to BLAST searches against our retrieved metagenomic sequences with the same threshold as above and were used as reference sequences in the phylogenetic analysis. Of the 117,066 retrieved sequences, 20,720 were use-restricted and were not considered further. The 96,345 remaining sequences were clustered at 90% sequence identity separately and combined into the final datasets. Sequences in the datasets of the 3 hallmark proteins were aligned using MAFFT v7 [96], followed by removal of poorly aligned positions by trimAl with gap threshold of 0.2 [97]. The phylogenetic trees were constructed using a JTT+CAT model by FastTree (version 2.1.11) [98], and visualized using ggtree v2.4.1 [99]. Biome information for IMG/M metagenomes were obtained from the Gold database [100]. All IMG/M contigs with a hit to an arTV marker were also cross-referenced with contigs in the IMG/VR database [101]. For contigs detected as viral and included in IMG/VR, the host prediction was obtained when available at the domain level (i.e., bacteria versus archaea) and displayed on the trees.

## Supporting information

**S1 Text. Supporting information text.**
(PDF)

**S1 Data. The dendrogram of CGJ distances for classified bacterial and archaeal dsDNA viruses.** CGJ, composite generalized Jaccard; dsDNA, double-stranded DNA.
(NWK)

**S2 Data. Raw data used to make graphs presented in this article.**
(XLSX)

**S1 Table. Properties of arTVs.** arTV, archaeal tailed virus.
(XLSX)

**S2 Table. PCs generated for arTVs by vConTACT v2.0.** arTV, archaeal tailed virus; PC, protein cluster.
(XLSX)

**S3 Table. Viral genome annotations.**
(XLSX)

**S4 Table. Virus-encoded DNA MTases.** MTase, methyltransferase.
(XLSX)

**S5 Table. Virus-encoded metabolic genes.**
(XLSX)

**S6 Table. Cobaltochelatase subunits gene cluster encoded by head-tailed viruses.**
(XLSX)

**S7 Table. Proviruses related to arTVs in archaeal genomes.** arTV, archaeal tailed virus.
(XLSX)

**S8 Table. Host ranges of haloarchaeal head-tailed viruses.**
(XLSX)

**S1 Fig. Genome alignment of arTVs.** Comparisons of genomes of **(A)** representative viruses from the 4 genera in *Hafunaviridae*, **(B)** viruses from the 2 genera in *Druskaviridae* and viruses

from the 2 genera in *Saparoviridae*, **(C)** HATV-2 from *Soleiviridae* and HGTV-1 from *Halomagnusviridae* (singletons), **(D)** the 3 viruses from *Haloferuviridae* (3 genera), **(E)** HATV-3 from *Pyrstoviridae*, HFTV1 from *Haloferuviridae*, HSTV-1 from *Shortaselviridae* and HRTV-28 from *Suolaviridae* (all singletons), **(F)** the 2 members from *Graaviviridae*, **(G)** the 3 viruses from *Vertoviridae* (2 genera), **(H)** psiM2 representing *Leisingerviridae* and Drs3 from *Anaerodiviridae*. Putative protein functions are indicated above or below the corresponding ORFs. Genes encoding virus morphogenesis–related proteins are colored in green, whereas replication-related genes are colored in red. Homologous genes shared between viruses are connected by shadings of different degrees of gray based on the amino acid sequence identity. See S1 Table for more information on families and genera. arTV, archaeal tailed virus.
(PDF)

**S2 Fig. The box plot shows the percentage of genes shared by arTVs.** The percentage of genes of a representative virus from each genus that shared with members of the proposed genus (G) and family (F), as well as arTVs from other families (A) are shown. Each box represents the middle 50th percentile of the data set and is derived using the lower and upper quartile values. The median value is displayed by a horizontal line. Whiskers represent the maximum and minimum values with the range of 1.5 IQR. Each virus is represented by dots. Proteins with over 30% amino acid sequence identity and E-value $< 1 \times 10^{-25}$ in our arTV database are counted as homologous proteins. Data underlying this figure can be found in S2 Data. arTV, archaeal tailed virus.
(PDF)

**S3 Fig. Multiple sequence alignments of different categories of MTases encoded by arTVs.** The boxed signature motifs are used to classify the MTases. arTV, archaeal tailed virus; MTase, methyltransferase.
(PDF)

**S4 Fig. Multiple sequence alignment of micrococcal nucleases encoded by arTVs.** The experimentally verified micrococcal nuclease encoded by *Staphylococcus hyicus* (AAA26661.1) is used as a sequence reference in the alignment. The empty and filled stars indicate the calcium binding and catalytic sites, respectively. arTV, archaeal tailed virus.
(PDF)

**S5 Fig. Multiple sequence alignment of putative L21e homologs encoded by arTVs from the family *Hafunaviridae*.** The L21e encoded by *Halorubrum californiensi*, *Natrinema pallidum*, and *Haloarcula marismortui* are used as references in the alignment. arTV, archaeal tailed virus.
(PDF)

**S6 Fig. Genome comparisons of the representative proviruses with related arTVs from 6 viral families.** Genes encoding virus morphogenesis proteins, genome replication proteins and integrases are colored in green, red and orange, respectively. For visualization purposes, genomes of some proviruses were circularized and reopened at a different position, matching the start site of the reference arTV genome. Homologous genes shared between (pro)viruses are connected by shadings of different degrees of gray based on the amino acid sequence identity. See S7 Table for complete lists of proviruses of *Hafunaviridae*, *Haloferuviridae*, *Graaviviridae*, *Vertoviridae*, *Leisingerviridae*, and *Anaerodiviridae*. arTV, archaeal tailed virus; *att*, attachment site.
(PDF)

**S7 Fig. The pairwise genome comparisons of viruses from the genera *Haloferacalesvirus* and *Mincapvirus* of the family *Hafunaviridae*.** Putative protein functions are indicted above the corresponding ORFs. Genes encoding virus morphogenesis related proteins are colored in green, whereas replication-related genes are in red. The putative tail fiber adhesin-encoding gene is shown in yellow. The genes encoding restriction modification enzymes are in blue. Homologous genes shared between viruses are connected by shadings of different degrees of gray based on the amino acid sequence identity.
(PDF)

**S8 Fig. The divergence of adhesin proteins encoded by viruses from the family *Hafunaviridae*.** **(A)** Pairwise sequence alignments of adhesins encoded by viruses from the genera *Haloferacalesvirus* and *Mincapvirus*. The HVSs, which are predicted to interact with host receptor, are indicated. **(B)** Phylogeny of the adhesins. Viruses that were used for EOP assay in this study are indicated in black, otherwise in gray. Nodes with bootstrap support values greater than 90% are indicated with dots. EOP, efficiency of plating; HVS, hypervariable segment.
(PDF)

**S9 Fig. The effect of mutations in adhesin to the host range changes in viruses of the 4 adhesin groups (see S8 Fig).** Pairwise sequence alignments were produced for each of the 4 adhesin groups. The HVSs identified in S8 Fig are indicated in each group. The ratio of shared numbers of sensitive hosts/the total numbers of sensitive hosts of each 2 tested viruses was calculated (see S2 Data), and the heat map of hs versus adhesin pi was generated for each adhesin group. hs, host range similarity; HVS, hypervariable segment; pi, protein identity.
(PDF)

**S10 Fig. Phylogenetic tree of VP1-like proteins encoded by viruses from the *Hafunaviridae*.** The same colors as in S8B Fig were used to indicate the Groups of adhesin proteins encoded by the corresponding viruses. Nodes with the bootstrap support values greater than 90% are indicated with dots.
(PDF)

**S11 Fig. Phylogenetic analysis of the portal and TerL proteins encoded by arTVs.** The homologs retrieved from the metagenomics databases were clustered at 90% identity, and the representative of each cluster is indicated as a dot in the branch with the color denoting the source of its biome. The circle outside of the tree consisting of tiles indicates the biome compositions from the corresponding PC, which is represented by the dot in the branch. The dot in the outermost ring indicates the (predicted) virus type (colored according to its putative host) in the corresponding branch. The names of the tailed virus isolates of haloarchaea and methanogens, as well as the fosmids generated from a saltern of Santa Pola in Spain, are shown on the tree. arTV, archaeal tailed virus; PC, protein cluster.
(PDF)

**S12 Fig. Maximum likelihood phylogenetic trees of the arTVs MCPs and portal proteins.** The best-fitting models identified for the MCP and portal trees were selected by PhyML and were Blosum62+G+I+F and LG+G+I+F, respectively. Same colors are used for clades containing the same group of viral members in both trees. HGTV-1 and related haloarchaeal virus fosmids, as well as ChaoS9, that clustered differently in the 2 trees are indicated by lines. Bootstrap values are shown next to the nodes, with values above 80 shown in black and those below 80 in gray. arTV, archaeal tailed virus; MCP, major capsid protein.
(PDF)

## Acknowledgments

We thank Sari Korhonen, Matti Ylänne, and Carlos Lapedriza for skillful technical assistance. We are also grateful to Marc Monot (Biomics Platform, Institut Pasteur) for help with the sequencing of viral genomes. The facilities and expertise of the HiLIFE Biocomplex unit at the University of Helsinki, a member of Instruct-ERIC Centre Finland, FINStruct, and Biocenter Finland are gratefully acknowledged.

## Author Contributions

**Conceptualization:** Ying Liu, Tatiana A. Demina, Hanna M. Oksanen, Mart Krupovic.

**Formal analysis:** Ying Liu, Tatiana A. Demina, Simon Roux, Pakorn Aiewsakun, Darius Kazlauskas.

**Funding acquisition:** David Prangishvili, Mart Krupovic.

**Methodology:** Pakorn Aiewsakun, Peter Simmonds.

**Resources:** Hanna M. Oksanen.

**Supervision:** Hanna M. Oksanen, Mart Krupovic.

**Visualization:** Ying Liu, Simon Roux.

**Writing – original draft:** Ying Liu, Mart Krupovic.

**Writing – review & editing:** Ying Liu, Tatiana A. Demina, Simon Roux, Pakorn Aiewsakun, Darius Kazlauskas, Peter Simmonds, David Prangishvili, Hanna M. Oksanen, Mart Krupovic.

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
