## [Editor Report · Decision Letter 0]

9 Jul 2021

Dear Dr. Krupovic, 

Thank you for submitting your manuscript entitled "Diversity, taxonomy and evolution of archaeal viruses of the class Caudoviricetes" for consideration as a Research Article by PLOS Biology.

Your manuscript has now been evaluated by the PLOS Biology editorial staff and I am writing to let you know that we would like to send your submission out for external peer review.

Please re-submit your manuscript within two working days, i.e. by Jul 11 2021 11:59PM.

Kind regards,

Paula

---

Paula Jauregui, PhD

Associate Editor

PLOS Biology

---

## [Decision Letter · Decision Letter 1]

8 Sep 2021

Dear Dr. Krupovic,

Thank you very much for submitting your manuscript "Diversity, taxonomy and evolution of archaeal viruses of the class Caudoviricetes" for consideration as a Research Article at PLOS Biology. Your manuscript has been evaluated by the PLOS Biology editors, an Academic Editor with relevant expertise, and by several independent reviewers.

In light of the reviews (below), we are pleased to offer you the opportunity to address the comments from the reviewers in a revised version that we anticipate should not take you very long. We will then assess your revised manuscript and your response to the reviewers' comments and we may consult the reviewers again.

In particular, reviewer #1 thinks you should add a more detailed analysis of the host range and the life cycle of the newly isolated viruses, asks whether the proposed viral families correlate with a similar host range, life cycle (integration or not) or sample location, whether some of the isolated viruses integrate, whether only viruses of a particular family encode integrases, and whether there are new conclusions (for example tRNAs enabling a broader host range). Reviewer #2 thinks that you should tone down a bit how robust the taxonomic framework is, provide more details and discussion of the arTV MCP and portal proteins, and include a brief discussion of how these arTVs are related or not related to other non-arTV archaeal viruses.

Please also address the following editorial requests:

DATA POLICY:

2) Deposition in a publicly available repository. **Please also provide the accession code or a reviewer link so that we may view your data before publication.**

Regardless of the method selected, please ensure that you provide the individual numerical values that underlie the summary data displayed in the following figure panels as they are essential for readers to assess your analysis and to reproduce it: Figures 3B, and 4A.

**Please also ensure that figure legends in your manuscript include information on where the underlying data can be found, and ensure your supplemental data file/s has a legend.**

**Please ensure that your Data Statement in the submission system accurately describes where your data can be found.**

Please also provide a blurb which (if accepted) will be included in out weekly and monthly Electronic Table of Contents, sent out to readers of PLOS Biology, and may be used to promote your article in social media. The blurb should be about 30-40 words long and is subject to editorial changes. It should, without exaggeration, entice people to read your manuscript. It should not be redundant with the title and should not contain acronyms or abbreviations.   

We expect to receive your revised manuscript within 1 month.

**IMPORTANT - SUBMITTING YOUR REVISION**

*Resubmission Checklist*

*Published Peer Review*

*PLOS Data Policy*

*Blot and Gel Data Policy*

Sincerely,

Paula

---

Paula Jauregui, PhD

Associate Editor

PLOS Biology

REVIEWS:

Reviewer #1: Archaeal viruses.

Reviewer #2: Archaeal viruses.

Reviewer #1: The manuscript entitled ‚Diversity, taxonomy and evolution of archaeal viruses of the class Caudoviricetes' is a comprehensive analysis of 37 new genomes of archaeal tailed viruses (arTVs) including already sequenced genomes of arTVs. The classification of the arTVs into 14 new viral families within the class Caudoviricetes is proposed based on analysis with GRAViTy and vConTACT.

Major comments:

The advantage of isolated viruses versus viral genomes from metagenomics data, in particular the possibility to determine the host range, is nicely pointed out in the introduction. The viruses sequenced here have been isolated in previous studies and the other virus genomes discussed here have been described in other publications. Therefore, a more detailed analysis of the host range and the life cycle of the newly isolated viruses would be of interest.

Questions that would be of interest:

Do the proposed viral families correlate with a similar host range, life cycle (integration or not), sample location?

Some of the isolated viruses encode integrases e.g. HRTV-8, HRTV-26, HRTV-27 - do they integrate? Integrases are not discussed when discussing the genomic content of viruses. Do only viruses of a particular family encode integrases?

Other comment:

Lane 290-297: The fact that HGTV-1 encodes a great number of tRNA is already discussed in earlier publications. Are there any new conclusions, for example tRNAs enabling a broader host range?

Minor comments:

A description of the supplementary tables would be nice in the supplementary file.

Supplementary table S9 Table description within the table:

Not clear: 'coloured fields have been tested' - 'empty fields have not been tested'. What about coloured empty fields - have they been tested - if yes what was the result, if not please clarify description.

Supplementary table S10 - spelling mistake lane 7 ('stains' instead of 'strains')

Supplementary information lane 155: 'encode' instead of 'encoding'

Main text lane 197-198: rephrase

Reviewer #2: The paper by Lie et al. provides new and valuable insights into the diversity, function, and phylogenetic relationships among the Caudoviricetes class of viruses. The work reported here more than doubles of knowledge of these arTV. It is remarkable that 63 arTV viral genomes can lead to the proposed formation of 14 new viral families. The phylogenetic analysis is well done, compelling, and supports the formation of these 14 new families. The presented (bioinformatic) annotation of genes in these new viruses is interesting and thoughtful without being too overly speculative. The data presenting on virus isolate's host range and their correlation with tail fiber adhesion proteins well done, although in retrospect, it is not all that surprising of a finding. More interesting was the (likely) different origins of some of the arTV viral families and their distinct separation from tailed bacteriophages. Overall, this is an excellent manuscript.

I have only minor suggestions for improving this manuscript. They include the following.

1. I would tone down a bit how robust the taxonomic framework is (i.e. lines 99-101) given that some families are represented by only a single member. As the author's themselves state, there is likely much more diversity out there in this class of viruses (lines 424-425).

2. It would be useful if the authors provide more details and discussion of the arTV MCP and portal proteins. How distinct or not are the secondary structures of the 9 MCP clades. Likewise for the portal proteins. Are the two trees coherent with each other or not?

3. The authors could also include a brief discussion of how these arTVs are related or not related to other non-arTV archaeal viruses.

---

## [Editor Report · Decision Letter 2]

17 Oct 2021

Dear Dr Krupovic,

On behalf of my colleagues and the Academic Editor, Curtis Suttle, I am pleased to say that we can in principle offer to publish your Research Article "Diversity, taxonomy and evolution of archaeal viruses of the class Caudoviricetes" in PLOS Biology, provided you address any remaining formatting and reporting issues. These will be detailed in an email that will follow this letter and that you will usually receive within 2-3 business days, during which time no action is required from you. Please note that we will not be able to formally accept your manuscript and schedule it for publication until you have made the required changes.

PRESS

Sincerely, 

Paula

---

Paula Jauregui, PhD 

Associate Editor 

PLOS Biology
